# Acute Inflammatory Response in Osteoporotic Fracture Healing Augmented with Mechanical Stimulation is Regulated In Vivo through the p38-MAPK Pathway

**DOI:** 10.3390/ijms22168720

**Published:** 2021-08-13

**Authors:** Simon Kwoon Ho Chow, Can Cui, Keith Yu Kin Cheng, Yu Ning Chim, Jinyu Wang, Carissa Hing Wai Wong, Ka Wai Ng, Ronald Man Yeung Wong, Wing Hoi Cheung

**Affiliations:** Musculoskeletal Research Laboratory, Department of Orthopaedics and Traumatology, The Chinese University of Hong Kong, Hong Kong; cancui@cuhk.edu.hk (C.C.); keithykcheng@link.cuhk.edu.hk (K.Y.K.C.); avicechim@link.cuhk.edu.hk (Y.N.C.); wangjinyu@link.cuhk.edu.hk (J.W.); hingwaicarissawong@cuhk.edu.hk (C.H.W.W.); kerry.ng@link.cuhk.edu.hk (K.W.N.); ronald.wong@cuhk.edu.hk (R.M.Y.W.)

**Keywords:** osteoporotic fracture healing, mechanical stimulation, inflammatory response, macrophage polarization

## Abstract

Low-magnitude high-frequency vibration (LMHFV) has previously been reported to modulate the acute inflammatory response of ovariectomy-induced osteoporotic fracture healing. However, the underlying mechanisms are not clear. In the present study, we investigated the effect of LMHFV on the inflammatory response and the role of the p38 MAPK mechanical signaling pathway in macrophages during the healing process. A closed femoral fracture SD rat model was used. In vivo results showed that LMHFV enhanced activation of the p38 MAPK pathway at the fracture site. The acute inflammatory response, expression of inflammatory cytokines, and callus formation were suppressed in vivo by p38 MAPK inhibition. However, LMHFV did not show direct in vitro enhancement effects on the polarization of RAW264.7 macrophage from the M1 to M2 phenotype, but instead promoted macrophage enlargement and transformation to dendritic monocytes. The present study demonstrated that p38 MAPK modulated the enhancement effects of mechanical stimulation in vivo only. LMHFV may not have exerted its enhancement effects directly on macrophage, but the exact mechanism may have taken a different pathway that requires further investigation in the various subsets of immune cells.

## 1. Introduction

Osteoporotic fracture is an emerging problem in older adults in the developed world and accounts for a high proportion of medical care costs worldwide. Fracture healing is a complex biological process that can coarsely be divided into three overlapping stages: inflammation, callus formation, and remodeling. The coordination of these processes determines the success of fracture healing. However, in osteoporotic fractures, recruitment of reparative mesenchymal stem cells (MSCs) [1], neo-angiogenesis at the callus [2], the level of endochondral ossification [3], rate of callus remodeling [4] and the expression of estrogen receptors (ERs) [5] were all impaired, leading to delayed healing or impaired healing outcomes.

The inflammatory stage is the first stage of fracture healing. When a fracture occurs, an inflammatory response is immediately casted to initiate and coordinate all the subsequent healing processes [6,7]. Different immune cells, such as macrophages, neutrophils, and lymphocytes, infiltrate the fracture site, and multiple inflammatory cytokines, including Tumor necrosis factor-α (TNF-α), Interleukin-6 (IL-6), and IL-10 are released [6]. This intense inflammatory interaction initiates the repair phase and ensures normal fracture healing through stimulating angiogenesis, attracting, and promoting differentiation of MSCs and enhancing extracellular matrix synthesis [8,9]. Any disruption of this inflammatory response could result in impaired or delayed fracture healing [10,11,12,13]. The early inflammatory response was shown to be disrupted after injury by the induction of estrogen-deficiency. In ovariectomized animals, inflammatory response was shown to be suboptimal at the fracture site as evidenced by decreased expression of pro-inflammatory cytokines, leading to inferior callus formation [14].

The p38 mitogen-activated protein kinases (p38 MAPK) is a member of the MAPK family, which controls many cellular processes. There is a large body of evidence showing that p38 MAPK activity is important for normal immune and inflammatory response. The p38 MAPK can be activated in macrophages, neutrophils, and T-cells by a number of extracellular mediators of inflammation, such as cytokines, chemokines, and chemoattractants [15]. In macrophages, p38 MAPK is involved in regulating the expressions of pro-inflammatory mediators, such as IL-1, TNF-α, Prostaglandin E2 (PGE2), and IL-12, thus playing a critical role in macrophage-mediated inflammation [16,17,18,19]. The expression of pro-inflammatory TNF-α in macrophages was shown to be regulated by the p38 MAPK pathway in a dose dependent manner [20]. Estradiol could activate p38 MAPKs in a time- and dose-dependent manner in bone marrow mesenchymal stem cells, and ovariectomy could lead to a decrease in p38 MAPK activation on bone sections [21].

At the same time, the p38 MAPK pathway has been shown to be essential for skeletogenesis and osteoblast differentiation [22,23,24]. Inhibition of p38 MAPK reduced alkaline phosphatase (ALP) expression in primary calvarial osteoblast cells, suggesting that activation of p38 MAPK is important for osteoblast differentiation [25]. Deletion of p38α in osteoblasts impaired the expression of Osterix (Osx), an osteoblast-specific transcription factor, resulting in defective osteoblast maturation and function [26]. Furthermore, p38 MAPK activation has been shown to be responsive to several types of mechanical loading, such as stretching [27], compressive force [28], and fluid shear [29], resulting in stimulated osteogenic differentiation. 

Low-magnitude high-frequency vibration (LMHFV) is a form of non-invasive, cyclic biophysical stimulation. Our previous studies demonstrated its beneficial effects on different fracture healing processes, including mesenchymal stem cell recruitment, neo-angiogenesis at the callus, endochondral ossification, and callus remodeling [2,3,4,30] in both normal and osteoporotic bones. This vibrational stimulation has also been shown to restore the compromised inflammatory response in OVX-induced osteoporotic fractures demonstrated by higher pro-inflammatory cytokine expressions and promoted polarization to reparative M2 phenotypes earlier at the fracture site [14,31]. However, mechanisms underlying this restoration process by LMHFV treatment are still unknown. 

Based on the evidence that p38 MAPK is critical for normal inflammatory response, especially in macrophage-mediated inflammation, and that P38 MAPK can be activated by mechanical loading, we hypothesize that the enhancement effect of LMHFV in osteoporotic fracture healing is via the p38 MAPK signaling pathway and the activation of p38 MAPK pathway in macrophages plays a key role in promoting osteogenesis. The objectives of this study were to examine the roles of the p38 MAPK pathway played in mechanical signal transduction in relation to the inflammatory response during osteoporotic fracture healing at tissue and cellular levels.

## 2. Results

### 2.1. LMHFV Enhanced Activation of p38 MAPK and the Inflammatory Response of Osteoporotic Fracture Healing

To investigate the activation of p38 MAPK at the fracture site, a relative p-p38 level was evaluated by normalizing p-p38 expression against total p38 expression by IHC. We found that the p-p38 level was markedly increased in the OVX-VT group at weeks 1 and 2 compared to the OVX group (Figure 1a, *p* = 0.029 and *p* = 0.000, respectively), suggesting that LMHFV was able to activate p38 MAPK at the fracture site in ovariectomized bones. A significantly lower p-p38 level was also detected in OVX+SB and OVX-VT+SB groups, compared to the OVX-VT group at weeks 1 and 2 (*p* ≤ 0.001 for both) and compared to the OVX group at week 2 (*p* = 0.004 and *p* = 0.008, respectively) (Figure 1a).

Locally, at the fracture site, LMHFV treatment enhanced the expressions of pro-inflammatory TNF-α and IL-6 at week 1 post-fracture (Figure 1b,c, *p* = 0.054, and *p* = 0.032, respectively) and suppressed anti-inflammatory IL-10 expression at weeks 1 and 2 (Figure 1d, *p* = 0.064 and *p* = 0.016, respectively) compared to the OVX group. Taken together, vibration treatment was found to enhance the local inflammatory response in OVX bones.

In terms of bone healing outcomes, vibration-treated animals showed better healing, as evidenced by radiographic images and microCT analysis. Serial X-ray images demonstrated that the OVX-VT group showed better callus formation and faster bridging of the callus gap compared to the OVX group at week 4 (Figure 2). Significantly higher CW and CA were observed in the OVX-VT group at weeks 1 and 2 (Figure 3). As for the microarchitecture of the callus, the OVX-VT group had higher Conn.D at week 1 (Figure 3, *p* = 0.000) and lower Tb.Sp at week 4 (Figure 4c, *p* = 0.023) compared to the OVX group. Therefore, our results re-confirmed that the vibration treatment had augmented osteoporotic fracture healing, along with an enhanced level of acute inflammatory response, which was accompanied with an enhanced activation of the p38-MAPK detected locally at the fracture site.

### 2.2. Inhibition of p38 MAPK Led to Impaired Inflammatory Response and Poorer Callus Formation

To confirm our results that LMHFV-induced phosphorylation of p38 MAPK was responsible for to the acute inflammatory response at early fracture healing, a specific p38 inhibitor (SB203580) was administered to the OVX rats. SB203580-treated animals showed lower p38 MAPK phosphorylation at the fracture site, as evidenced by significantly decreased p-p38 positive signals in the inhibitor groups (OVX-SB and OVX-VT-SB) at weeks 1 and 2 (Figure 1a). Compared to OVX-VT group at weeks 1 and 2, significantly decreased TNF-α expressions were found in OVX-SB (*p* = 0.02 and *p* = 0.01, respectively) and OVX-VT-SB groups (*p* = 0.000 for both) (Figure 1b). The OVX-VT-SB group also showed lower IL-6 expression than the OVX-VT group at weeks 1 and 2 (Figure 1c). Systematically, all inhibitor groups showed lower inflammatory cytokine levels in serum. TNF-α, IL-6, and IL-10 levels were markedly downregulated upon SB203580 treatment (Figure 5).

Regarding the callus formation capacity, the inhibitor groups showed significantly lower CW and CA compared to the OVX and OVX-VT groups at weeks 1 and 2 (Figure 3). The OVX-VT group had significantly lower CA than the OVX-SB group at week 8. Three-dimensional images by MicroCT demonstrated that callus gaps were still apparent in the OVX, OVX-SB, and OVX-VT-SB groups at week 8 (Figure 4). The OVX-SB group showed significantly lower TV compared to the OVX-VT group 1 at weeks 1 and 2. Conn.D was found to be lower in the two inhibitor groups than the OVX-VT group. Significantly higher Tb.Sp was detected in the OVX-SB group rather than the OVX-VT group at weeks 1 and 2 (Figure 4c, *p* = 0.025 and *p* = 0.029, respectively). Taken together, the inhibition of p38 MAPK activity could result in lower inflammatory response both locally and systemically in the early healing stage and poorer callus formation capacity and properties (Figure 5). The treatment effects of vibration therapy in terms of the enhancements in inflammatory response and callus formation were impeded by the inhibition of p38 MAPK.

### 2.3. LMHFV Did Not Directly Enhance Macrophage Polarization In Vitro 

To further confirm the role of p38 MAPK during the enhancement by LMHFV treatment on fracture healing, we investigated the effect of LMHFV on monocyte/macrophages and their polarization (Figure 6). 

LPS treatment showed strong intracytoplasmic labeling of M1 for inducible nitric oxide synthase (iNOS) (Figure 6a). LPS-treated M1 macrophages are characterized by an enlarged amoeboid cell shape with roundish cell bodies and numerous delicate cytoplasmic extensions on the cellular surface. The LPS+SB group, LPS+VT group, and LPS+VT+SB group showed larger M1 macrophage cell size, more multinucleid fused cells, yet fewer total cell numbers. Immunofluorescent staining of iNOS showed that LPS treatment significantly increased the iNOS positive area percentage compared with the CTL group (*p* < 0.001), and iNOS expression by IF staining was lower in macrophage cells treated with the p38 MAPK inhibitor (SB and VT+SB samples). The SB group, VT group, and SB+VT group significantly decreased the iNOS positive area percentage compared with the LPS group (all at *p* < 0.0001). The iNOS positive area percentage of the VT group was also significantly lower than the CTL group (*p* < 0.01) and SB group (*p* < 0.05). Macrophage cells treated with LPS evaluated by ELISA showed that the SB group significantly lowered TNF- α expression compared with both the CTL group and VT group (both at *p* < 0.05), while the VT+SB group showed significantly lower TNF-a concentrations compared with both the CTL group and VT group (*p* < 0.05 and *p* < 0.01) and a lower trend of the IL-6 concentration than the CTL group and VT group (*p* = 0.06 and *p* = 0.07). The cells treated with SB203580 had markedly suppressed levels of the pro-inflammatory cytokines, TNF-α and IL-6. Therefore, vibration treatment was not shown to have directly enhanced the transformation of macrophages to the M1 phenotype in vitro.

A morphological transformation of RAW264.7 cells from macrophage-like cells into dendritic-like cells was clearly observed at the CTL group after 24 h incubation with LPS. IL-4-treated RAW264.7 macrophages demonstrated larger multi-nuclei, prominent nucleoli, and relatively prominent cytoplasm with increased granularity. Large “spindeloid” macrophages with an elongated cell body and cytoplasmic extensions on the apical ends of the cell bodies and numerous multinucleated giant cells (MNGs) with abundant cytoplasmic projections on the cellular surface were present in the IL-4 group. M2 cells in the IL4+SB group, IL4+VT group, and IL4+VT+SB group underwent migration and adhesion via filopodial projections or fusopods and formed binucleated cells. Intense membranous expression of CD209 antigen by M2-macrophages was observed among all treatments. The IL4 group showed a higher trend of F4/80 area compared with the CTL group (*p* = 0.07). The IL4+VT group showed a significantly lower ratio of M2-CD209 to M1-F4/80 area than the CTL group (*p* < 0.05). The IL4+VT group showed a significantly lower F4/80 area than the IL4 group (*p* < 0.0001) and IL4+SB group (*p* < 0.05), while the CD209 area in the IL4+VT group was also lower than the IL4 group (*p* < 0.0001) and IL4+SB group (*p* < 0.05). The IL4+VT+SB group showed lower F4/80 area (*p* = 0.06) and significantly lower CD209 area compared with the IL4 group (*p* < 0.05). IL4 stimulation could significantly increase IL-10 concentration compared with the SB group (*p* < 0.01) and SB+VT group (*p* < 0.05). The VT group also showed significantly higher IL-10 concentration than the SB group (*p* < 0.01) and SB+VT group (*p* < 0.05). Altogether, vibration treatment did not show a direct effect on the polarization of M1 to M2 phenotypes in vitro.

## 3. Discussion

This study focused on the potential roles of the p38 MAPK signaling pathway in LMHFV-augmented osteoporotic fracture healing. p38 MAPK has been reported to play important roles in inflammatory response, as well as bone development and remodeling, but its role in fracture healing has not been well understood. Our in vivo results confirmed that LMHFV treatment can restore the impaired inflammatory response and enhance callus formation in OVX-induced osteoporotic fracture healing. The restoration of inflammatory response and enhancement in callus formation are modulated by the p38 MAPK pathway, which can be activated by LMHFV. However, LMHFV did not show significant effects on macrophage polarization in vitro but only suppressed by the inhibition of p38 MAPK. 

In the present study, LMHFV was able to activate p38 MAPK at the fracture site in OVX bones, as evidenced by a significantly higher relative p-p38 level in the OVX-VT group compared to the OVX group at weeks 1 and 2. Estrogen is known to be involved in the activation of the p38 MAPK signaling pathway [32,33], and OVX was previously found to diminish p38 MAPK activation on bone sections in mice [21]. Previous reports revealed that vibration loading could activate the p38 MAPK pathway in bone marrow-derived stem cells (BMSCs) [34,35]. However, there is no known report on p38 MAPK activation in fracture healing with or without mechanical stimulation. Our finding demonstrated that vibration treatment could enhance p38 MAPK activation at the fracture site in osteoporotic fracture healing.

The current study once again demonstrated that LMHFV could promote the expressions of pro-inflammatory cytokines of TNF-α and IL-6, suppress the anti-inflammatory cytokine of IL-10 expression, and increase the M2 surface marker, CD206, expression locally at the fracture site in OVX bone at week 1. The application of vibration treatment had no influence on the inflammatory response systemically (Figure 5). These findings are consistent with our previous study on the inflammatory response in fracture healing [14]. p38 MAPK is well-known for playing important roles in the inflammatory response and is involved in the production of pro-inflammatory cytokines, such as TNF-α and IL-1 [36,37,38]. Upon phosphorylation, p38 MAPK has shown to activate transcription factors regulating a variety of target gene expressions, including pro-inflammatory cytokines [38]. It is also been reported that p38 MAPK is essential for bone formation and enhance osteogenic differentiation when being activated in MSCs [39,40]. 

In vivo inhibition of p38 MAPK by SB203580 abolished the enhancement effects by LMHFV on inflammatory response and callus formation. Our findings indicated that the inhibition of p38 MAPK activity by SB203580 could lead to impaired inflammatory response in terms of suppressed cytokine levels, both locally and systemically. In line with our findings, TNF-α expression was previously shown to be reduced in rat pancreatic tissues with the administration of SB203580 via IP injection [41]. These findings suggested that the p38 MAPK pathway plays an important role in the acute inflammatory response and previously reported macrophage activation [14] and cytokine expression during fracture healing. Our findings in callus formation and remodeling are also consistent and well supported by a number of in vitro models. Lu et al. and Xiao et al. reported that blocking the p38 MAPK signaling pathway by SB203580 inhibited osteogenic differentiation of BMSCs induced by mechanical stretching [27] and vibration loading [35]. Xu et al. demonstrated that inhibition of p38 MAPK reduced (bone morphogenetic protein) BMP9-induced ALP activity and calcium deposition in MSCs [42]. Thus, together with our in vivo data, these findings underlined the importance of p38 MAPK activity in osteogenesis. Overall, the inhibition of p38 MAPK resulted in the suppression of vibration treatment effects in terms of inflammatory response and callus formation in osteoporotic fracture healing that is regulated in vivo via the activation of the p38 MAPK pathway.

Inflammatory responses are largely mediated by macrophages. There is a growing amount of evidence suggesting that p38 MAPK are essential protein kinases in macrophage-mediated inflammatory responses. For example, p38α is involved in the expression of pro-inflammatory mediators in macrophages such as IL-1β, TNF-α, and PGE2 [19]. Thus, our present study focused on the role of p38 MAPK in macrophages, which also plays a key role in fracture healing. In contrary to our in vivo data, LMHFV was not shown to have enhanced the transformation of macrophage polarization from the M0 to M1, M1 to M2 polarization. Our finding is not completely in line with previous studies, reporting that macrophage morphology and activation could be modulated by mechanical stimuli [43] and that mechanical stretch was shown to induce alternative activation of macrophages (M2) to facilitate hair regeneration in mouse models [44], probably due to the form of stimulation given. Instead, we have observed that vibration has positively substantiated the transformation macrophages into dendritic-like cells with an enlarged cell size that is induced by LPS (Figure 6a). Although LMHFV did not directly enhance the polarization of M1 to M2, as shown by the surface markers of F4/80+ and CD209+ signals, these cell morphology changes are accompanied with increased pro-inflammatory cytokine expression (TNF-α and IL-6, Figure 6c) and increased anti-inflammatory cytokine expression (IL-10, Figure 7c), which suggested that vibration was promoting the RAW264.7 cells to differentiate into a specific subset of dendritic monocytes/macrophages [45] with a distinctive function during the inflammatory stage in fracture healing that could not be easily defined into M1 or M2. As dendritic monocytes are required for the activation of T-cells in bone injury and regeneration [46] and T-cells and B-cells are required for the accumulation of bone forming cells [47], our results suggest that LMHFV may be exerting its promotive effects on the acute inflammatory response during fracture healing by promoting monocyte precursors to differentiate into a subset of immune cells, including dendritic monocytes for the activation of T-cells and the subsequent accumulation of bone forming cells. This postulation would require further experiments to investigate. 

Several limitations exist in the present study; only p38 MAPK in macrophages and its interactions with mechanical stimulation were evaluated. Hence, further studies should be conducted to investigate other immune cells, bone cells, and signaling pathways that may also play key roles in mechanical signal transduction during fracture healing. The well-established closed femoral fracture model selected in this study, except healing processes, showed degrees of endochondral and intramembranous ossifications, and the response to mechanical stimulation may vary between different regions and affect the amount of callus formed. Thus, other fracture models, such as the more clinically relevant metaphyseal fracture model [48,49], can be used.

In conclusion, our study has demonstrated that the in vivo enhancement of inflammatory response by LMHFV treatment was only modulated in vivo by p38 MAPK. LMHFV treatment promoted in vitro the transformation of RAW264.7 macrophages to a subset of cells resembling dendritic monocytes, which is independent of the p38-MAPK pathway. 

## 4. Materials and Methods

### 4.1. Animal Model and Interventions

In total, 96 (*n* = 96) female Sprague-Dawley rats were used in this study. They were supplied by the Laboratory Animal Services Centre of the Chinese University of Hong Kong, and the experimental protocols were approved by the University Animal Experimentation Ethics Committee (Ref: 17-175-MIS). All rats were housed at the Research Animal Laboratory in the authors’ institution, with 12-h light–night cycle, and they were given free cage movement with access to standard rat chow and tap water ad libitum. 

All rats were subject to bilateral ovariectomy (OVX) operations at 6 months of age. In brief, under general anesthesia, OVX surgery was performed by making bilateral incisions at the dorsal lower abdomen, followed by the removal of ovaries. Animals were aged for 3 months for development of osteoporosis, as previously described [50]. Nine-month-old rats were randomly divided into four groups: (1) ovariectomized (OVX); (2) ovariectomized control with the administration of SB203580, a specific p38 MAPK inhibitor (OVX-SB); (3) ovariectomized vibration (OVX-VT); and (4) ovariectomized vibration with the administration of SB203580 (OVX-VT-SB). SB203580 is a pyridinyl-imidazole inhibitor used to inhibit the catalytic activity of p38 MAPK by binding competitively at the ATP-binding pocket and has been used widely to study p38 MAPK functions [51,52,53]. 

All rats received closed fracture creation on the right femoral midshaft based on our established protocol [1,30]. Briefly, a 1.2 mm diameter sterilized Kirschner wire was inserted into the medullary canal, followed by fracture creation using a customized 3-point bending apparatus. In total, 1 mg/kg [54] of SB203580 (Selleckchem, Houston, TX, USA) was administered to the inhibitor groups through intraperitoneal (IP) injection daily, starting from the day after fracture creation. The OVX-VT and OVX-VT-SB groups received LMHFV (35 Hz, 0.3 g; g = gravitational acceleration) treatment starting from 2 days after fracture creation. The treatment was given daily for 20 min/day and 5 days/week, according to our previous protocol [2,55]. The OVX and OVX-SB groups were put on the vibration platform with power off. Six rats from each group were euthanized using an overdose of intraperitoneal (IP) injection of sodium pentobarbital at 200 mg/kg at weeks 1, 2, 4, and 8 post-treatment, and the femora were harvested for micro-CT and histomorphometric assessments.

### 4.2. Radiographic Analysis

To monitor the fracture healing status, weekly radiographies of the rat femora were taken with a cabinet X-ray system (UltraFocus DXA, Faxitron, Lincolnshire, IL, USA). Callus width (CW) and area (CA) were measured on digitized lateral view images using the built-in straight line and polygon selection tools in image analysis software ImageJ (NIH, Baltimore, MA, USA). CW was defined as the maximal outer diameter of the mineralized callus minus the outer diameter of the femur; CA was calculated as the sum of the areas of the external mineralized callus [30].

### 4.3. 3-D Bone Morphometry and Micro-Finite Element Analysis

At euthanasia, the fractured femora were harvested and scanned with a micro-CT system (VivaCT 40, Scanco Medical, Brüttisellen, Switzerland) to quantify changes in bone mineral density (BMD) and microarchitecture. In total, 422 slides covering 8.02 mm proximal and distal to the fracture line were scanned and defined as the region of interest (ROI) [1,5,30]. A threshold set at 220–1000 was used to distinguish between non-mineralized and mineralized tissue [14]. Tissues within this set range were evaluated and reconstructed. Total tissue volume (TV), total bone volume (BV), bone volume fraction (BV/TV), and BMD of TV and BV were assessed. Stiffness, apparent modulus, and failure load at week 8 were estimated by the micro-finite element analysis (micro-FEA, FE-software version 1.13, Scanco Medical, Brüttisellen, Switzerland) by the axial compression test in the z-direction with the material modulus defined at 10 GPa using the default evaluation program, as previously reported [56].

### 4.4. Systemic Inflammatory Cytokine Levels

Before euthanasia, a whole blood sample was collected through cardiac puncture and allowed to conjugate at room temperature for 2 h, followed by centrifugation at 1000× *g* for 20 min for serum isolation. Serum samples were stored at −80 °C until further analysis. Serum TNF-α, IL-6, and IL-10 concentrations were measured using enzyme-linked immunosorbent assay (ELISA). Commercial kits (RTA00, R6000B, R1000; Quantikine^®^ ELISA; R&D Systems, Minneapolis, MN, USA) were utilized according to the manufacturer’s instructions. Optical density of each well was determined using a microplate reader (µQuant™, BioTek^®^ Instruments, Winooski, VT, USA).

### 4.5. Bone Histomorphometry

After microCT scanning, the femora were fixed in 10% neutral buffered formalin and decalcified in 9% formic acid. Processed specimens were paraffin embedded and sectioned into 5-µm-thick slices along the sagittal plane of the femora. The sections were then stained with haematoxylin–eosin (H&E) (Sigma-Aldrich, St. Louis, MO, USA) to demonstrate the callus morphometry histologically [30,55].

### 4.6. Immunohistochemistry (IHC)

Histological sections were subject to antigen retrieval in 60 °C citrate buffer. Primary antibodies against TNF-α (1:100; ab6671; Abcam, Cambridge, UK), IL-6 (1:200; NB600-1131; Novus Biologicals, Littleton, Colorado, USA), IL-10 (1:100; ARC0102, Invitrogen, Waltham, MA, USA), total p38 (1:200; NBP2-19662; Novus Biologicals), and phosphorylated p38 (p-p38) (1:100; GTX59567; GeneTex, Eching, Germany) were applied to the sections and incubated overnight at 4 °C. For negative control, primary antibodies were replaced by an isotope control antibody (IgG; GeneTex). All other steps followed the manufacturer’s instructions (ab236469; Abcam), and all specimens were processed following the identical procedures. Finally, the sections were counterstained by hematoxylin and images were captured on a Leica microscope system (DMRXA2, Leica Microsystems GmbH, Wetzlar, Germany). Quantitative analysis of the positively stained area was performed at the bony callus and compared with the negative control. Expressions of the target protein were quantified by color threshold in ImageJ [5].

### 4.7. Macrophage RAW 264.7 Culture

Murine RAW264.7 macrophage cells were cultured in Dulbecco’s Modified Eagle’s Medium Nutrient Mixture F-12 (DMEM/F12) (Gibco), supplemented with 10% fetal bovine serum (FBS) (Gibco) and 1% Penicillin-Streptomycin-Neomycin (PSN) Antibiotic Mixture (Gibco), at 37 °C in a humidified 5% CO_2_ atmosphere. The cells were divided into four groups (*n* = 3 per group): control group with no treatment (CTL), inhibitor group (SB), vibration group (VT), and vibration plus inhibitor group (VT+SB). LMHFV treatment was given to the vibration groups, as described above, and SB203580 (20µM) was applied to the inhibitor groups. Macrophage cells were seeded in 6-well plates at a density of 1 × 10^6^ cells per well and polarized to M1 by 12-h exposure to 50 ng/mL lipopolysaccharide (LPS; Sigma-Aldrich) in macrophage medium. LMHFV treatment was applied to the cells 1 h after the addition of LPS. Polarized M1 macrophages were treated for 1 h with or without SB203580 and then stimulated with IL-4 (20 ng/mL) (Murine IL-4; PreproTech, Rocky Hill, NJ, USA) for 12 h. LMHFV treatment was applied to the cells 1 h after the addition of IL-4. The polarization technique with LPS, and IL-4 has been previously demonstrated by flow cytometry, quantitative real-time polymerase chain reaction (qRT-PCR), and cytokine secretion profile to produce M1 and M2 phenotypes reliably [57,58,59]. Conditioned medium (CM) was collected and centrifuged at 1000× *g* for 20 min at 4 °C to remove floating cells and cell debris and frozen at −80 °C until use. 

### 4.8. In Vitro Inflammatory Cytokine Levels

Culture supernatants from M1 polarized RAW264.7 were collected and examined for concentrations of TNF-α and IL-6 by ELISA kits (DY410-05, DY406-05, R&D Systems), following the manufacturer’s instructions. As for IL-10 concentration measurement (M1000B, R&D Systems), another batch of M1 macrophages polarized from RAW264.7 were treated, as described above. Absorbance was measured at a wavelength of 450 nm using a microplate reader, as described above. All standards and samples were assayed in duplicate.

### 4.9. Immunofluorescence Assay for M1/M2 Markers

RAW 264.7 macrophages were seeded at the density of 5 × 10^5^ cells per well over coverslips in 6-well plates, activated as described above. Cells were fixed for 30 min with PBS containing 4% formaldehyde and washed three times with PBS. Non-specific binding was blocked by treatment of cells with bovine serum albumin (BSA, 4%; Thermo Fisher, Waltham, MA, USA), diluted in PBS for 1 h at room temperature. M1 cells were incubated with iNOS rabbit Polyclonal antibody (NB300-605, 1:200, Novus Biologicals) at 4 °C overnight, and M2 cells were incubated with F4/80 mouse monoclonal antibody, Alexa Fluor 488 (1:200, Catlog #53-4801-82, Thermo Fisher) and CD209 Rabbit mAb (1:200, A9649, ABclonal, Woburn, MA, USA) at 4 °C overnight. Secondary antibodies included goat anti-mouse IgG1 cross-adsorbed secondary antibody, Alexa Fluor 488 (1:500, Catalog# A-21121, Thermo Fisher), and goat anti-rabbit IgG (H+L) cross-adsorbed secondary antibody, Alexa Fluor 594 (1:500, Catalog# A11037, Thermo Fisher). Coverslips containing cells were mounted with the Prolong Gold antifade reagent with DAPI (Life Technologies, Carlsbad, CA, USA), visualized under a fluorescence microscope (Leica, DM 6000B, Werzlar, Germany) at 400× magnifications, and analyzed with Image J (NIH, Bethesda, MD, USA).

### 4.10. Statistical Analysis

All quantitative data were presented as mean ± standard deviation and analyzed in SPSS 25.0 software (IBM, NY, USA). Normality of data were confirmed by the Kolmogorov-Smirnov test. All data were analyzed using one-way analysis of variance (ANOVA) to evaluate differences among groups, separately at different time points, followed by post-hoc Bonferroni’s multiple comparison tests. All statistical significance was considered at *p* ≤ 0.05.

## Figures and Tables

**Figure 1 ijms-22-08720-f001:**
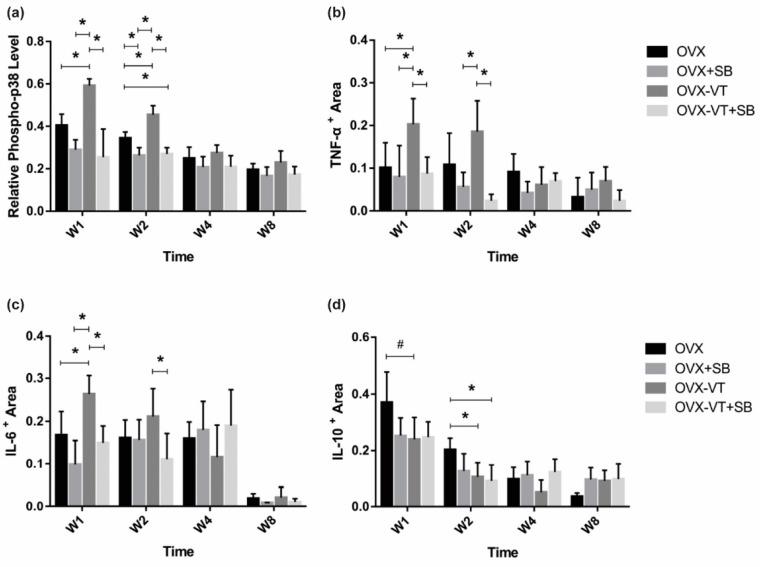
Immunohistochemistry results. Relative expression of major inflammatory cytokines were quantified by immunohistochemistry at the fracture callus. (**a**) Relative pp-38 was found to be significantly increased by VT in OVX animals, and the inhibition of p38 removed the VT treatment effect in week 1 post fracture; (**b**) pro-inflammatory TNF-α showed a similar pattern of expression; (**c**) as well as the pro-inflammatory cytokine IL-6, (**d**) the anti-inflammatory cytokine IL-10 showed decreased expression after the SB, VT, and VT+SB treatment at weeks 1 and 2. * Statistical significance considered at *p* < 0.05, # approaching significance at *p* = 0.064, one-way ANOVA with post-hoc Bonferroni test. Abbreviation: W: week.

**Figure 2 ijms-22-08720-f002:**
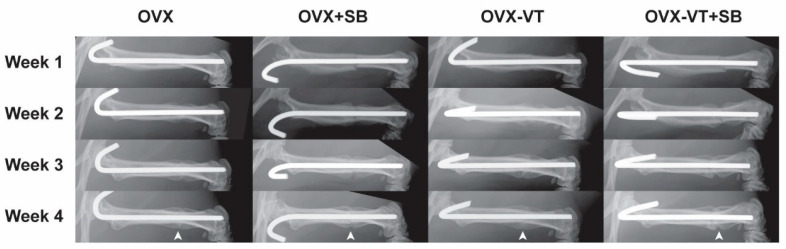
Representative x-ray images of the four groups from week 1 to week 4 during the early inflammatory stage to overlapping anabolic callus formation stage.

**Figure 3 ijms-22-08720-f003:**
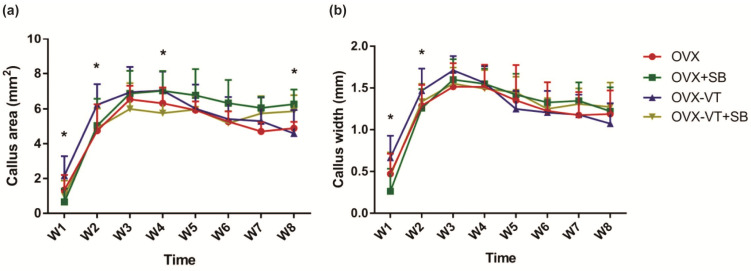
Histomorphometry of callus size. (**a**) VT enhanced the callus area as soon as weeks 1 and 2, with statistically significant difference, and lasted until week 4; (**b**) Callus width was higher in the OVX-VT treatment group compared to the other group in weeks 1 and 2. * Statistical significance considered at *p* < 0.05, one-way ANOVA with post-hoc Bonferroni test. Abbreviation: W: week.

**Figure 4 ijms-22-08720-f004:**
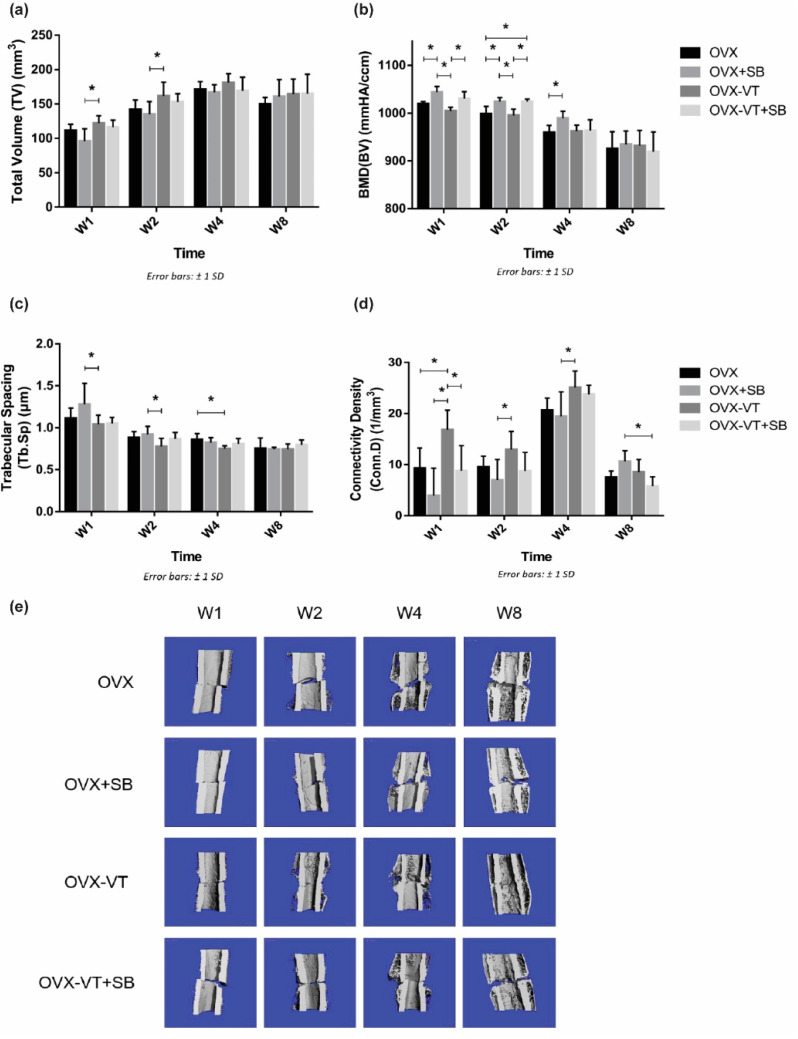
MicroCT evaluation of fracture site. (**a**) Tissue volume (TV) was shown to be enhanced by the vibration treatment at weeks 1 and 2 compared to other groups. (**b**) Segmented Bone volume (BV) was found to be enhanced by the VT at weeks 1 and 2, and the enhancement effects were abolished when p-38 was inhibited with a specific inhibitor. (**c**) Trabecular spacing (Tb.Sp) was lower in the vibration treated group from week 1 to 4. (**d**) Connectivity density (Conn.D) was found to be higher in the vibration treated group from weeks 1 to 4, and p-38 inhibition removed all treatment effects. (**e**) Representative microCT 3D-reconstructed images showing all groups at each end-point. * Statistical significance considered at *p* < 0.05, one-way ANOVA with post-hoc Bonferroni test. Abbreviation: W: week.

**Figure 5 ijms-22-08720-f005:**
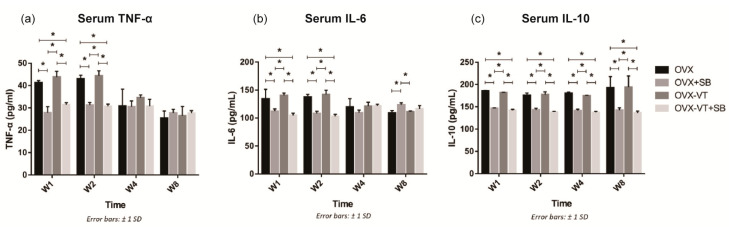
Systemic inflammatory cytokine levels. (**a**) Significantly higher TNF-α levels were detected in the OVX and OVX-VT groups compared to the OVX+SB and OVX-VT+SB groups at weeks 1 and 2. (**b**) The OVX and OVX-VT groups showed significantly higher IL-6 level than the OVX+SB and OVX-VT+SB groups at weeks 1 and 2 (* *p* < 0.0005 for all). The OVX+SB group demonstrated higher IL-6 level compared to the OVX and OVX-VT groups at week 8. (**c**) Significantly higher IL-10 levels were detected in the OVX and OVX-VT groups compared to the OVX+SB and OVX-VT+SB groups at all time points. * *p* < 0.0005 for all, one-way ANOVA with post-hoc Bonferroni test. Abbreviation: W: week.

**Figure 6 ijms-22-08720-f006:**
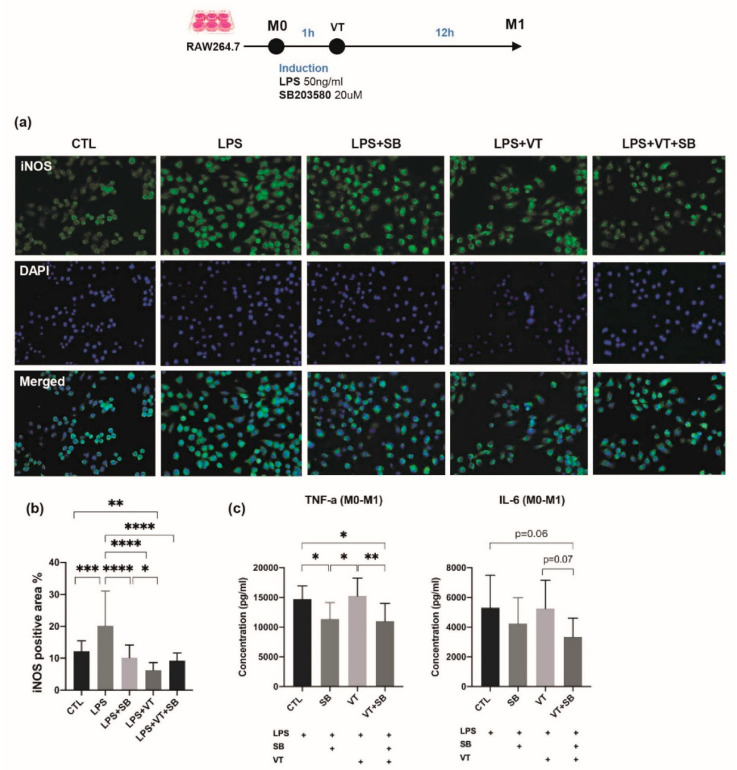
Immunofluorescent staining and inflammatory cytokine levels among different treatment groups for induction of M1 macrophage. (**a**) Immunofluorescence staining of in vitro cultured RAW 264.7 macrophages labeled with the M1 marker of inducible nitric oxide synthase (iNOS) and nuclear counterstaining with DAPI. LPS treatment showed strong intracytoplasmic labeling of M1 for iNOS. LPS-treated M1 macrophages were characterized by an enlarged amoeboid cell shape with roundish cell bodies and numerous delicate cytoplasmic extensions on the cellular surface. The LPS+SB group, LPS+VT group, and LPS+VT+SB group showed larger cell sizes, more multinucleid-fused cells, yet fewer cell numbers. (**b**) LPS treatment significantly increased the iNOS positive area percentage compared with the CTL group, while the SB group, VT group, and SB+VT group significantly decreased the iNOS positive area percentage compared with the LPS group. The percentage of the iNOS positive area of the VT group was also significantly lower than the CTL group and SB group. (**c**) The SB group showed significantly lower TNF-α concentrations compared with both the CTL group and VT group, while the VT+SB group showed significantly lower TNF-α concentrations compared with both the CTL group and VT group and lower trend of IL-6 concentration than the CTL group and VT group (*n* = 10). * *p* < 0.05, ** *p* < 0.01, *** *p* < 0.001, and **** *p* < 0.0001. Scale bars = 50 μm.

**Figure 7 ijms-22-08720-f007:**
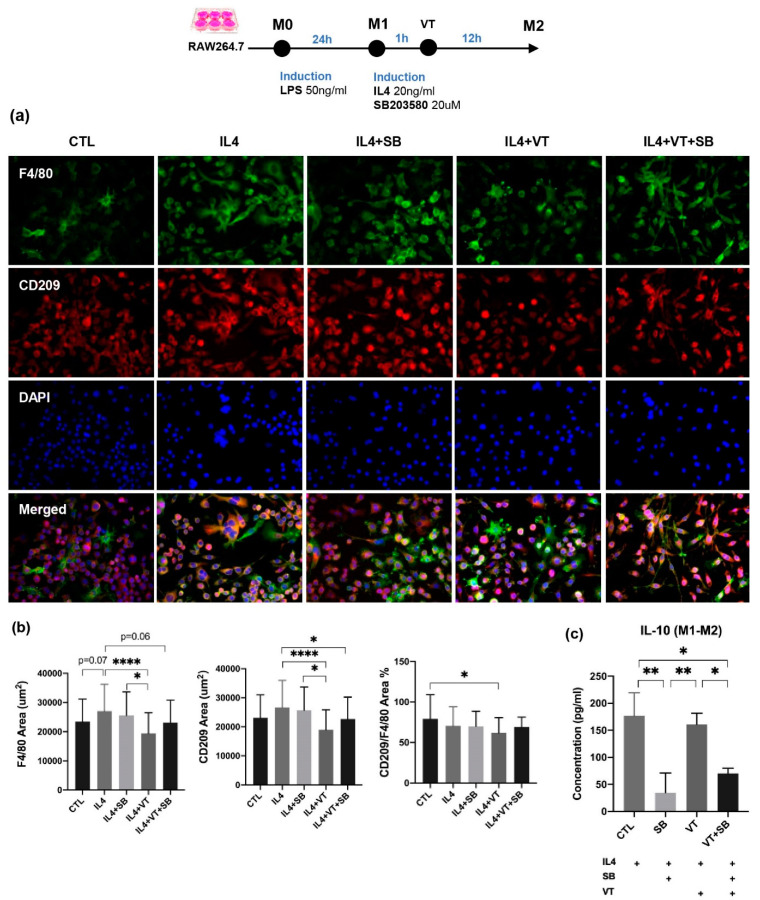
Immunofluorescent staining and inflammatory cytokine levels among different treatment groups for M2 macrophage. (**a**) Immunofluorescence staining of in vitro cultured RAW 264.7 macrophages labeled with F4/80 and CD209 and nuclear counterstaining with DAPI. A morphological transformation of RAW264.7 cells from macrophage-like cells into dendritic-like cells was clearly observed in the CTL group after 24 h incubation with LPS. IL-4-treated RAW264.7 cells acquired larger multi-nuclei, prominent nucleoli, and relatively prominent cytoplasm with increased granularity. Large “spindeloid” macrophages with an elongated cell body and cytoplasmic extensions on the apical ends of the cell bodies and numerous multinucleated giant cells (MNGs) with abundant cytoplasmic projections on the cellular surface were present in the IL-4 group. M2 cells in the IL4+SB group, IL4+VT group, and IL4+VT+SB group underwent migration and adhesion via filopodial projections or fusopods and formed binucleated cells. Intense membranous expression of CD209 antigen by M2-macrophages were observed among all treatments. (**b**) The IL4 group showed a higher trend of F4/80 area compared with the CTL group. The IL4+VT group showed a significantly lower CD209/F4/80 area percentage than the CTL group. The IL4+VT group showed a significantly lower F4/80 area and CD209 area than the IL4 group and IL4+SB group. The IL4+VT+SB group showed a lower F4/80 area and significantly lower CD209 area compared with the IL4 group. (**c**) All groups were treated with IL4. The CTL group (with IL4 stimulation) showed significantly higher IL-10 concentration than the SB group and SB+VT group. The VT group (with IL4 stimulation) showed significantly higher IL-10 concentration than the SB group and SB+VT group (*n* = 3). * *p* < 0.05, ** *p* < 0.01, and **** *p* < 0.0001. Scale bars = 50 μm.

## Data Availability

The data presented in this study are available on request from the corresponding author. The data are not publicly available due to reasons related institutional data policies.

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
