# Peer review of "Acute Inflammatory Response in Osteoporotic Fracture Healing Augmented with Mechanical Stimulation is Regulated In Vivo through the p38-MAPK Pathway"

_ijms, 2021, doi:10.3390/ijms22168720_

Round 1
Reviewer 1 Report
Did you consider that the healing process could differ in the different fracture stabilization and different bone position. Long axis positioning could increase ossification. Did you consider to use different animal model. Please ad it to limitations.
Author Response
- Did you consider that the healing process could differ in the different fracture stabilization and different bone position? Long axis positioning could increase ossification. Did you consider to use different animal model. Please add it to limitations.
Thank you for the suggestion and the point regarding amount of callus formed in various fracture site and fracture type has been supplemented in the limitation section in line 340 to 341. We have also proposed the potential difference between endochondral versus intramembraneous ossification in diaphyseal and metaphyseal fractures with literature support.
Reviewer 2 Report
The authors studied the molecular mechanisms of LMHFV effects on osteoporotic fracture healing. They proved that LMHFV effects in vivo are regulated by p38 MAPK pathway and associated with initial increase of inflammation followed by faster callus formation. Moreover, the authors’ in vitro studies demonstrated association of LMHFV treatment with RAW264.7 macrophage enlargement and transformation to dendritic monocytes presumably capable of activating T-cells and accumulation of bone-forming cells. The study is very important.
Comments
- The Title of the paper should be modified, as macrophage polarization usually involving M1-M2 formation was not observed.
- Lines 18-20: This sentence is not clear it should be rephrased.
- Lines 92-93: The authors should specify which object (animals, cells, etc.?) was used in this set of experiments.
- Figs 1, 3, 4, 5: W1…W8 abbreviations should be disclosed in the Figure legends.
- Fig 2 is missing. This should be corrected.
- Lines 291-293: The authors did not study in vivo the effect of MAPK pathway on macrophage activation; therefore, they should modify their suggestion.
Author Response
- The Title of the paper should be modified, as macrophage polarization usually involving M1-M2 formation was not observed.
Thank you for your comment. It is well taken and we have revised the title to reflect better our observations. The new title is proposed to be “Acute inflammatory response in osteoporotic fracture healing augmented with mechanical stimulation is regulated in vivo through the p38-MAPK pathway”.
Macrophage polarization was previously observed by using M1 marker of iNOS and M2 marker of CD206 utilizing the same osteoporotic fracture rat model with similar expression pattern of pro-inflammatory and anti-inflammatory cytokine. We previously reported the observation [Chow et al, Eur Cell Mater 38:228-45].
- Lines 18-20: This sentence is not clear it should be rephrased.
Thank you for pointing out the flaw, the sentence has been revised to better deliver the message.
- Lines 92-93: The authors should specify which object (animals, cells, etc.?) was used in this set of experiments.
This set of experiment was reported from the “osteoporotic animal model” at in the sub-heading of the paragraph. IHC data was also reported at the “fracture site” as in line 92.
- Figs 1, 3, 4, 5: W1…W8 abbreviations should be disclosed in the Figure legends.
Abbreviation key is added back to each of the figure legends.
- Fig 2 is missing. This should be corrected.
The figure is re-inserted into the manuscript, please kindly check if it displays properly.
- Lines 291-293: The authors did not study in vivo the effect of MAPK pathway on macrophage activation; therefore, they should modify their suggestion.
Thank you for the comment. The suggested was indeed drawn from current and previously reported observations. Relevant literature support has been added and the statement is revised accordingly at line 294 to 295.